# Risk Factors Associated with Opportunistic Infections among People Living with HIV/AIDS and Receiving an Antiretroviral Therapy in Gabon, Central Africa

**DOI:** 10.3390/v16010085

**Published:** 2024-01-04

**Authors:** Augustin Mouinga-Ondeme, Neil Michel Longo-Pendy, Ivan Cyr Moussadji Kinga, Barthélémy Ngoubangoye, Pamela Moussavou-Boundzanga, Larson Boundenga, Abdoulaye Diane, Jeanne Sica, Ivan Sosthene Mfouo-Tynga, Edgard Brice Ngoungou

**Affiliations:** 1Unité des Infections Rétrovirales et Pathologies Associées, Centre Interdisciplinaire de Recherches Médicales de Franceville (CIRMF), Franceville BP 769, Gabon; pamelamoussavoub@gmail.com (P.M.-B.); dianeyabdoulaye@gmail.com (A.D.); tivansdavids2012@gmail.com (I.S.M.-T.); 2Unité de Recherches en Ecologie de la Santé, Centre Interdisciplinaire de Recherches Médicales de Franceville (CIRMF), Franceville BP 769, Gabon; longo2michel@gmail.com (N.M.L.-P.); boundenga@gmail.com (L.B.); 3Centre de Primatologie, Centre Interdisciplinaire de Recherches Médicales de Franceville (CIRMF), Franceville BP 769, Gabon; cyrth01@yahoo.fr (I.C.M.K.); genistha@hotmail.com (B.N.); 4Département d’Anthropologie, Université de Durham, South Road, Durham DH1 3LE, UK; 5Centre de Traitement Ambulatoire, Franceville BP 277, Gabon; jeannesic@yahoo.fr; 6Département d’Epidémiologie, Biostatistiques et Informatique Médicale (DEBIM)/Unité de Recherche en Epidémiologie des Maladies Chroniques et Santé Environnement (UREMCSE), Faculté de Médecine, Université des Sciences de la Santé, Libreville-Owendo BP 18231, Gabon; ngoungou2001@yahoo.fr

**Keywords:** HIV-1, PLWHA, opportunistic infections, Gabon

## Abstract

The Human Immunodeficiency Virus/Acquired Immunodeficiency Syndrome (HIV/AIDS) is still one of the main causes of death in sub-Saharan Africa. Antiretroviral therapies (ARTs) have significantly improved the health conditions of people living with HIV/AIDS (PLWHA). Consequently, a significant drop in morbidity and mortality, along with a reduced incidence of opportunistic infections (OIs), has been observed. However, certain atypical and biological profiles emerge in ART patients post-examination. The objective of this study was to identify the risk factors that contributed to the onset of OIs in HIV patients undergoing ART in Gabon. Epidemiological and biological data were obtained from medical records (2017 to 2019) found at the outpatient treatment centre (CTA) of Franceville in Gabon. Samples for blood count, CD4, and viral load analysis at CIRMF were collected from PLWHA suffering from other pathogen-induced conditions. A survey was carried out and data were analysed using Rstudio 4.0.2 and Excel 2007 software. Biological and socio-demographic characteristics were examined concerning OIs through both a univariate analysis via Fisher’s exact tests or chi^2^ (χ^2^), and a multivariate analysis via logistic regression. Out of the 300 participants initially selected, 223 were included in the study, including 154 (69.05%) women and 69 (30.95%) men. The mean age was 40 (38.6; 41.85), with individuals ranging from 2 to 77 years old. The study cohort was classified into five age groups (2 to 12, 20 to 29, 30 to 39, 40 to 49, and 50 to 77 years old), among which the groups aged 30 to 39 and 40 to 49 emerged as the largest, comprising 68 (30.5%) and 75 (33.6%) participants, respectively. It was noted that 57.9% of PLWHA had developed OIs and three subgroups were distinguished, with parasitic, viral, and bacterial infections present in 18%, 39.7%, and 55.4% of cases, respectively. There was a correlation between being male and having a low CD4 T-cell count and the onset of OIs. The study revealed a high overall prevalence of OIs, and extending the study to other regions of Gabon would yield a better understanding of the risk factors associated with the onset of these infections.

## 1. Introduction

The vulgarization of antiretroviral therapy (ART) has improved the living conditions of people living with HIV/AIDS (PLWHA) and has led to a significant decrease in morbidity and mortality [1,2,3]. Factors related to ART—late initiation, discontinuation, poor adherence, low CD4 T lymphocyte count, inadequate virological monitoring, isoniazid preventive therapy, gender, age, place of residence, and functional or disclosure or nutritional status—have been associated with the emergence of opportunistic infections (OIs) among adults (PLWHA) post-ART [4,5,6,7]. Frequently, OIs occur at all stages of AIDS, leading to increased gravity of the condition and worse health status of PLWHA [8]. This constitutes a major cause of hospitalization, and subsequent death in patients, highlighting the issue of management [9]. The prevalence of OIs among ART-treated adults varies and depends on the specificity of each country; for example, it was estimated at 22.4% in the south-east of Nigeria, from 33.6 to 88.44% in Ethiopia, and at 33.51% in seven provinces of Indonesia [10,11]. Some of the commonly described OIs in PLWHA are salmonella infection, chronic diarrhoea, candidiasis, toxoplasmosis, tuberculosis, dermatitis, and pneumonia [1].

At the end of 2022, the World Health Organization (WHO) estimated that 39 million people were HIV-infected and 650,000 died from AIDS-related illnesses worldwide; the fatalities were mainly due to OIs [12]. During the last Demographic and Health Survey (DHS), HIV-1 infection was reported in 4.1% of the general population aged 15 to 49 years old in Gabon [13]. The distribution of HIV-1 across the country showed that the Haut-Ogooué province, in the south-eastern part of the country, was one of the most affected regions, with 4.2% of HIV-1 infected people, and around 54% of PLWHA receiving ART in Gabon [13]. Treatment administration in this country follows WHO recommendations for resource-limited settings/countries and includes generalized first-line ART regimens: Recommended first-line regimens until 2019 included two nucleoside reverse transcriptase inhibitors plus one non-nucleoside reverse transcriptase inhibitor (2NRTI + 1 NNRTI) [14]. Due to mutations associated with drug resistance, WHO recommended a transition to integrase strand transfer inhibitor (INSTI)-based first-line regimens [15]. Studies have been conducted on HIV in co-infection with other pathogens in Gabon, but there is only one study, conducted in Libreville (capital city), highlighting OIs among PLWHA [16]. No study on OIs was found concerning PLWHA in rural or semi-rural areas, including the Haut-Ogooué province of Gabon. Recently, we reported a high prevalence of HTLV-1/HIV-1 co-infection among PLWHA followed-up at the outpatient or ambulatory treatment centre (CTA) of Franceville, the major city of Haut-Ogooué province, Gabon [17]. The need for preventive strategies to effectively manage the scenarios of co-infections in PLWHA was clearly identified and recommended.

Our intent was to assess the incidence and associated risk factors of OIs, so we considered the HIV-1-infected patients receiving ART monitoring at the CTA of Franceville. In this sanitary facility, specific care has been provided for several HIV/AIDS patients of all ages, both male and female, including pregnant women, from the entire Haut-Ogooué province.

## 2. Materials and Methods

### 2.1. Study Design and Sampling

From January to June 2020, we conducted a retrospective and cross-sectional study of PLWHA who were monitored at the CTA in Franceville, a city of roughly 7000 inhabitants in the south-eastern region of Gabon. Patients were recruited between June 2018 and September 2019 to perform a one-time CD4 count and HIV-1 RNA viral load measurement at diagnosis (as described in Section 2.3) free of cost to patients at the Centre Interdisciplinaire de Recherches Médicales de Franceville (CIRMF). According to medical records (2017–2019) filed at the CTA of Franceville, the recruitment criteria were as follows: (i) being confirmed HIV-1-positive, (ii) being on antiretroviral therapy (ART), and (iii) providing informed consent. The additional information collected included gender, age, ART regimen, and CD4 count at ART initiation. Patients who did not show a CD4 count and/or viral load during diagnosis of the opportunistic diseases, and those who had interrupted their follow-up at the CTA, were excluded from the study. Figure 1 presents the timeline of the study.

### 2.2. Ethical Consideration

The study was approved by the Gabon National Ethics Committee for Research (approval registered as PROT N°0011/2013/SG/CNE) and all participants gave verbal consent before participation in the study.

### 2.3. Sampling for HIV Viral Load and CD4 Counts

For determining HIV viral load (VL) and CD4 counts, blood specimens were collected from each patient using 5 mL EDTA tubes and transferred to a reference national laboratory, located at the Unit of Retroviral Infection and Associated Pathology, CIRMF. On arrival at the facility, samples were centrifuged, and then plasma and buffy coat fractions were collected and stored at −80 °C until testing, as previously described [17]. Specifically, CD4 counts at enrolment were determined using flow cytometry (Fascount, Becton Dickinson, San Jose, CA, USA). Plasma HIV-1 VL was determined using the Generic HIV Viral Load test (Biocentric, Bandol, France) performed according to the manufacturer’s instructions, using 200 µL of plasma and the protocol with a detection limit of 300 copies/mL [17,18].

### 2.4. Diagnosis of Other Pathogenic (Viral, Parasitic, and Bacteriological) Infections

The search for opportunistic infections (OIs) was carried out in 2020 in patients initially recruited between 2017 and 2019 (as shown in Figure 1). Plasma samples from patients were screened for hepatitis B virus infection (HBV surface antigen [19]) by simultaneously using a rapid diagnosis test (RDT) (AgHBs 2 DETERMINE™, Abbott Rapid Diagnostics Ltd, Stockport, UK) and an automated quantitative test (ù). Similarly, the screening for hepatitis C virus (HCV) infection was based on the antibodies anti-HCV by simultaneously using an RDT (Accu-Tell^®^ HCV Rapid Test Cassette/Strip (Serum/Plasma), Accubiotech, Beijing, China) and an automated quantitative test (VIDAS^®^ Anti-HCV (HCV), bioMérieux, Marcy-L’Etoile, France). For parasitic infection, toxoplasmosis serological analysis was performed with an automated quantitative test (VIDAS^®^ Toxo IgG II and VIDAS^®^ ToxoIgM kits, bioMérieux, Marcy-L’Etoile, France). The presence of IgG meant that the organism showed an acquired immunity with respect to Toxoplasma, while the presence of IgM indicated a sign of acute Toxoplasma infection. For bacteriological diagnosis, Chlamydia infection was screened via serological diagnosis, which evaluated the presence of IgG and IgM specific to Chlamydia, by using CHLAMYCHECK IgG and IgA. On the other hand, the “treponemal test” and the automated quantitative Elecsys^®^ Syphilis for cobas E411 (Roche Diagnostics, Mannheim, Germany) were used for the diagnosis of syphilis. In case of positivity, a second test called the “non-treponemal test” (VDRL assay with the SIMPLI RPR CARBO kit from MEDIFF°, Aubagne, France) was performed to determine the status of infection (active or not). The direct diagnosis of gonorrhoea infection was performed via urethral cytological examination for male and endo-cervical or vaginal for female samples. An attempt to determine the presence of Gram-negative Cocci in diplococci “coffee bean”, was performed using a bacteriological culture of genital samples on VCA3 medium [20]. Finally, tuberculosis (TB) diagnosis was performed by using chest X-rays, skin tests, and sputum examinations in a specialized care centre.

### 2.5. Occurrence of Opportunistic Infections

In order to emphasize the co-occurrence between HIV infection and OIs, a Venn diagram was generated using the “Venn Diagram” package. Furthermore, a binomial generalized linear model (GLM) was performed using the “stats” package to investigate the effect of seven factors as explanatory variables in the GLM, namely immunological (CD4 T lymphocyte count), virological (viral load), gender (sex), age, marital status, level of ART participation/consistency/lost to follow-up, and geographical origin, on the likelihood of OIs in PLWHA, which serves as the dependent variable in the model [21]. The GLM holds that any observed response is a linear sum of multiple individuals’ underlying responses. The GLM was used to determine how the predetermined factors (7) might have combined and/or contributed to the occurrence of OIs.

### 2.6. Statistical Analysis

The information collected was recorded in an Excel file and analysis was conducted with R software version 4.0.2 (http://cran.r-project.org/bin/windows/base/old/4.0.2/NEWS.R-4.0.2.html, accessed on 22 June 2020). The interaction level among variables (factors) was estimated with the Akaike information criterion (AIC) for model selection, and the AIC changes were analysed by utilizing the dredge function within the MuMIn package [21]. As a consequence of the AIC change analysis, only the variables (two factors) that had an impact on the incidence of OIs were chosen and included in the model, out of the original seven explanatory variables. The model with the lowest AIC was considered the best, while ΔAIC values below 2 were also retained [22]. Subsequently, we conducted the chi^2^ significance test (*p* < 0.05) to compare the proportional variance among the various forms of OIs.

## 3. Results

### 3.1. Characteristics of Patients

Of the 300 patients selected, 223 were included in the study; the participants had a median age of 41 (IQR 27.5–54.5) and their ages ranged from 2 to 77 years old, comprising 69 (30.9%) males and 154 (69.1%) females. The study cohort was categorised into five age groups (2 to 12, 20 to 29, 30 to 39, 40 to 49, and 50 to 77 years old) with the two most populated age groups (30 to 39 and 40 to 49) emerging as prominent, comprising 68 (30.5%) and 75 (33.6%) participants, respectively. Participants had their CD4 T-cell counts available upon inclusion, and the median count was determined to be 419 (93–745). The median duration of ART was 20 months, and all participants were taking ART as a first line of treatment. Two ART regimens were the most prevalent, consisting of zidovudine (AZT) plus lamivudine (3TC) along with efavirenz (EFV)/nevirapine (NVP) (25.8%), and tenofovir (TDF) plus 3TC/emtricitabine (FTC) plus EFV (15%) (Table 1).

### 3.2. Prevalence, Distribution, and Spectrum of Opportunistic Infections

Approximately 57.9% of PLWHA were diagnosed with OIs, and pulmonary TB (55.4%) was the most prevalent OI after ART initiation. This condition is considered as the primary bacterial infection. The viral (HBV and HCV) infections (39.7%) and parasitic infections (mainly oral candidiasis; 18%), ranked second and third, respectively (Table 1).

### 3.3. Different Combinations of Opportunistic Infections among People Living with HIV

The Venn diagram (Figure 2) and proportions comparison indicate that the populace consists of HIV-infected patients linked with other OIs (57.9%) against HIV-mono-infected patients ((42.1%) (Chi^2^ = 10.368, *p*-value = 0.001)). In OIs in HIV bi-infections, bacterial conditions illnesses were more frequent, followed by virus-induced ones (Chi^2^ = 78.624, *p*-value < 0.0001). Interestingly, the comorbidity made up of tri-infections of HIV, bacteria, and parasites was the most prevalent, with only two individuals in the studied population being tetra-infected with HIV, bacteria, virus, and parasites (Figure 2).

### 3.4. Risk Factors Associated with the Occurrence of Opportunistic Infections

The incidence of OIs was higher among men (68.1%) than women (53.2%). Moreover, the age group consisting of 20–29-year-old individuals had the highest OI prevalence rate, estimated at 70.4%. Participants with a CD4 T-cell count value of [200–499] and <200 cells/mm^3^ exhibited a greater likelihood of developing OIs than those with higher CD4 T-cell count values (value ≥ 500 cells/mm^3^). Similarly, a relatively high incidence of OIs was observed in patients who were administered TDF-3TC-EFV as their ART regimen (61.5%), closely followed by TDF + FTC + EFV and AZT-3TC-EFV, with an estimated rate of 57.4% for both.

The best model (ΔAIC = 0) resulting from our generalized linear model (GLM) indicates that two out of the seven sociobiological factors identified among PLWHA receiving ART were linked to the development of OIs. These factors were identified as gender and the initial value of the CD4 T-cell count (Table 2 and Figure 3). Consequently, male patients (Figure 3A) with suboptimal CD4 lymphocyte levels (Figure 3B) had a greater risk of developing OIs.

## 4. Discussion

In this study, we describe opportunistic infections (OIs) among people living with HIV/AIDS (PLWHA) in the south-east region of Gabon, with the majority of participants consisting of women, as reported in previous studies on this population. Most HIV-1 infected patients receiving ART in that CTA in Gabon are women, and there is no new finding in this regard [17,18,23]. Previous studies conducted in other countries on a similar type (female vs. male) of population have also described that women were the predominant gender [24,25]. We found that 57.9% of PLWHA had developed OIs, of which 18% were parasitic, 39.7% viral, and 55.4% bacterial infections. Being a male with a low CD4 T-cell count are the main joint-risk factors associated with the development of OIs.

The same population has been considered to study other pathogen-induced infections, including hepatitis viruses (HCV), and human T-lymphotropic virus type 1 (HTLV-1) [17]. The prevalence rate of 57.9% for OIs found among infected people confirms the vulnerability of PLWHA in Gabon with a high risk of exposure to viral, parasitic, and bacteriological agents. This reported rate is higher than those previously described in Gabon [16,26]. There is regularly a shortage of ARVs experienced in Gabon, and that might assuredly constitute a significant contributor and possible explanation for the high rates of OIs. Furthermore, starting ART at later stages of HIV infection, on the one hand, and the disparity observed in the level of engagement in HIV care facilities in poor countries or/and resource-limited areas, on the other hand, limit the effectiveness of the administered therapy [1,23]. Additionally, the emerging or increasing levels of HIV drug resistance to NNRTIs in resource-limited countries could also be considered as restraining factors [15]. At the time of the study, all participants were receiving the recommended first-line regimens, which included 2 NRTI + 1 NNRTI in Gabon. The obtained prevalence rate of OIs is comparable to the 55.3% that was reported among HIV/AIDS patients on ART in the southern zone Tigray, and the 60.97% rate in the western part of Ethiopia [24,27]. However, it is higher than the prevalence reported in similar studies conducted in other regions of Ethiopia, and in Nigeria and Taiwan [10,28,29,30,31,32]. In addition, our findings are lower than the 83.81% and 88.44% rates reported among HIV/AIDS patients in other distinctive parts of Ethiopia [5].

Analysis of bacteriological, viral, and parasitic OIs found that pulmonary TB, HBV, HCV, and oral candidiasis were the commonly occurring OIs in the studied population of Franceville. Bacteriological infections represented only by pulmonary TB (55.4%) were the most prevalent OI among PLWHA. Viral infections with HBV and HCV (39.7%) and parasitic infections with oral candidiasis (18%) ranked second and third, respectively. This finding about pulmonary TB in our study was congruent with another study conducted in Libreville, the Capital city of Gabon, where this OI was reported as the major cause of morbidity [16]. Other previous studies conducted in Gabon showed that the prevalence rate of TB/HIV co-infection was estimated at 26% to 42% [33,34,35,36]. Pulmonary TB is considered a leading cause of death in PLWHA, and having the highest prevalence reported in the current study highlights the risk related to this vulnerable population progressing rapidly towards the AIDS stage and developing further complications [37]. These findings emphasize the need for more surveillance and delicate care of PLWHA.

Viral OIs including HBV and HCV were the second most prevalent and induced diseases in the current study with a rate of 39.7%. Both hepatitis viruses are prevalent in Gabon and have already been described in co-infection with HIV [19,38,39]. The HIV/HCV and HIV/HBV co-infections could result from unprotected sexual intercourse; these diseases have been shown to be transmitted through unprotected sex with infected partners [40]. As in mono-infection, viral hepatitis B- and C-related diseases were estimated to be responsible for a significantly high number of deaths [41]. HBV and HCV infections can cause acute and chronic hepatitis, leading to liver fibrosis, cirrhosis, hepatocellular carcinoma (HCC), and end-stage liver disease over time [42]. In the era of ART, these diseases are considered to account for an increased probability of deaths among PLWHA [43,44,45]. Therefore, continuous surveillance must be conducted among PLWHA to clarify the current status of hepatitis viral diseases as well as parasitic infections. Parasitic infection, or oral candidiasis (18%), is one of the main OIs, together with the above-mentioned ones. Generally, oral candidiasis is prevalent among PLWHA, as previously described in Gabon (14.19%), as well as reported in India and Ethiopia [16,25,46]. The prevalence rate reported in the present study was higher than that previously reported in Gabon (14.19% vs. 18%). All these findings could be explained by the difference between a relatively rural area (current study) and more urbanized settlements. This coincided with previous studies from Ethiopia and the USA, where it was demonstrated that rural/countryside patients with lower levels of education, low income, and limited access to healthcare facilities more readily develop OIs than others [25,47,48,49].

Our study showed that being a male with a low CD4 T-cell count constitutes a higher risk that is associated with the development of OIs. We have already indicated that most HIV-1 infected patients and monitored ART recipients at most care centres (CTA) in Gabon are women [17,18,23]. Despite receiving ART, OIs were reported in the current study. The fact that men are more likely to develop OIs could be explained by the decision to adhere to and receive ART at the latter stages or/and the poor adherence and subsequent side effects [1]. It is recognised that, unlike women who often attend healthcare facilities as quickly as possible when they get ill/diagnosed, their male counterparts are prompt to seek medical advice when facing health-related complications or severe cases [50]. Information collected from participants showed that most men unfortunately do not accept the reality of being infected with HIV/AIDS, and so delay taking and adhering to recommended actions. Not only do they refuse or delay ARTs or/and take them discontinuously, but sometimes they continue to engage in inappropriate conduct such as unprotected sex. The obvious consequence is the emergence of HIV-associated complications, and sometimes mutations with drug resistance reported in Africa, as it is mainly observed in Gabon [14,18]. In Gabon, ART guidelines at the time of this study were based on WHO recommendations for resource-limited settings/countries and included generalized first-line ART regimens: Recommended first-line regimens included 2 NRTI + 1 NNRTI [14]. Due to mutations associated with drug resistance, WHO recommended the transition to integrase strand transfer inhibitor (INSTI)-based first-line regimens [15]. The high prevalence of OIs among patients receiving TDF-3TC/FTC+EFV was probably also due to possible mutations associated with resistance. And the low CD4 T-cell count could be interrelated with treatment failure. CD4 T-cell counts of [200–499] and <200 cells/mm^3^ were reported in our study, and similar counts are most likely reported in cases of development of OIs. This result is reasonable as CD4 T-cells play critical roles in the induction of both humoral and cellular immunity to combat OIs [1]. Comparatively, other authors found that PLWHA with a CD4 T-cell count of 200 cells/mm^3^ were at higher risk of developing advanced forms of OIs such as pulmonary TB [51]. Also, previous studies described that there are patients who develop OIs despite the high CD4 T-cell count [52,53]. This finding was in concordance with our results, showing that the risk of developing OIs is not only due to low CD4 T-cell count. In fact, a high CD4 T-cell count could be an abnormal increase in case of co-infection between HIV-1 and HTLV-1 conferring false immunity to the patients, as recently reported in a similar population type [17].

One limitation of this study was the relatively small number of participants, in comparison to the entire active patient database of the CTA, with a significant number of “lost to follow-up” individuals, and thus not consistently monitored. Furthermore, the study was solely conducted in a semi-rural location of a single CTA, making it difficult to evaluate how the obtained results might translate to urban areas or other regions of the country with PLWHA. Hence, as this study was retrospective, certain characteristics of the patients that were not documented, including their level of knowledge, marital status, and professional category or geographical location could influence and skew the analysis of the results. Therefore, the necessity of improving the management of PLWHA will enhance treatment outcomes. In the future, it will be critically important to consider all aspects of OIs, drug resistance, patient behaviour regarding treatment compliance, and attendance at CTAs when devising strategies for the care of PLWHA in Gabon. Poor compliance or medication abandonment may also heighten the likelihood of OIs. When developing such approaches, the involvement of personnel such as health staff, researchers, social workers, politicians, and traditional practitioners should be considered as all of these people may influence and have a significant impact on the healthcare of those specific groups of PLWHA.

## 5. Conclusions

Our study reports a significant occurrence of opportunistic infections (OIs) amongst HIV-1-infected patients receiving first-line ART and being monitored at CTA of Franceville in the south-east part of Gabon. We examine the range and risk factors related to the development of these OIs. The major OIs observed in the study were pulmonary TB, hepatitis B and C, and oral candidiasis. Additionally, the development of OIs was linked to male gender and low CD4 T-cell count. It is imperative to devise strategies to tackle the burden of OIs and improve outcomes, specifically for those with severely weakened immune systems. Our research emphasises the necessity of offering enhanced monitoring and care for HIV-positive individuals. Furthermore, carrying out extensive investigations in various provinces across the country will aid in identifying and comprehending all OIs that are prevalent, ultimately enabling the development of national strategies to mitigate their impacts on the health of PLWHA.

## Figures and Tables

**Figure 1 viruses-16-00085-f001:**
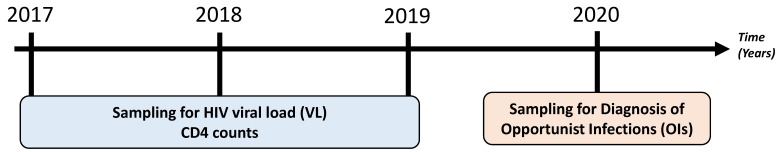
Schematic representation of the timeline of the study. Progress in carrying out examinations on the people living with HIV/AIDS (PLWHA) throughout the study period.

**Figure 2 viruses-16-00085-f002:**
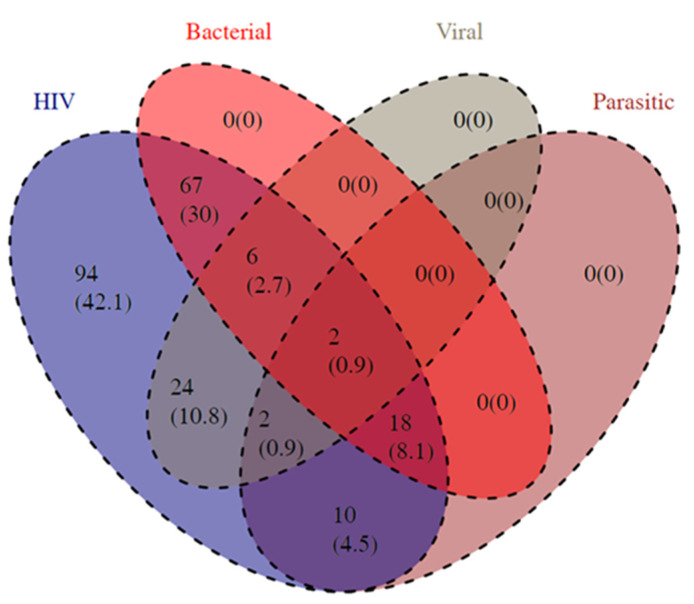
Different combinations of opportunistic infections among people living with HIV. The Venn diagram illustrates the proportions of individuals infected with HIV only (blue), those afflicted with bacterial infections only (red), those with viral infections only (grey), and those diagnosed with parasitic infections only (indigo/purple). The mixed colours indicated the proportions of PLWHA co-infected with HIV and one OI group, two OI groups, three OI groups, or different combinations between OI groups. The resulting and merged intermediate colours reflect the proportions and combinations observed.

**Figure 3 viruses-16-00085-f003:**
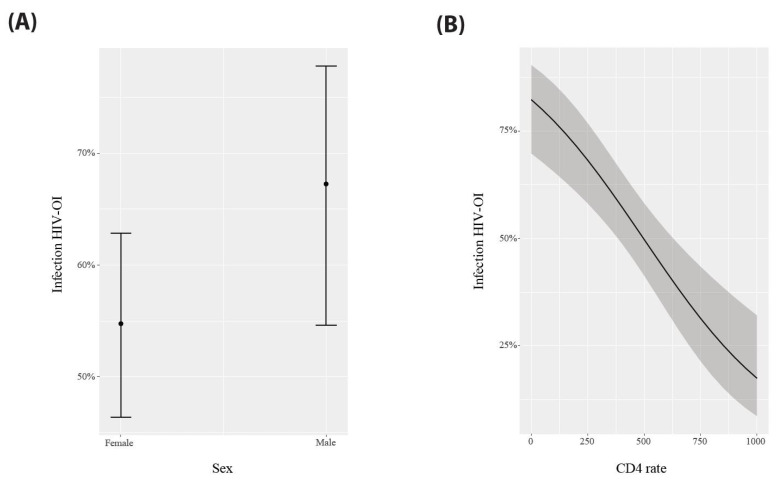
Probability of occurrence of opportunistic infections among people living with HIV. The combined effect of being male (**A**) with a low CD4 T-cell count (**B**) appears to be the most critical joint-risk factor associated with the development of opportunistic infections (OIs).

**Table 1 viruses-16-00085-t001:** Patient characteristics associated with opportunistic infections.

Variables		OIs Prevalence	Virological OIs	Bacteriological OIs	Parasitic OIs
Gender	n (%)	n/N (%)	n (% [95% CI])	n (% [95% CI])	n (% [95% CI])
Men	69 (30.9)	47/69 (68.1)	11 (16 [8.6–27.2])	35 (50.7 [38.5–62.9])	11 (16 [8.6–27.2])
Women	154 (69.1)	82/154 (53.2)	23 (15 [9.9-21.8])	58 (37.7 [30.1–45.9])	21 (13.6 [8.8–20.3])
Total	223 (100)	129/223 (57.9)	34 (15 [11–20.8])	93 (41.7 [35.2–48.5])	32 (14.3 [10.2–19.8])
Age (Years)					
2–12	9(4.1)	6/9 (66.7)	1 (11 [0.6–49.3])	4 (44.4 [15.3–77.3])	3 (33.3 [9–69.1])
20–29	27(12.1)	19/27 (70.4)	7 (26 [11.9–46.6])	13 (48.1 [29.2–67.6])	4 (14.8 [4.9–34.6])
30–39	68(30.5)	35/68 (51.5)	14 (20.6 [12.1–32.5	19 (28 [18.1–40.3])	6 (8.8 [3.6–18.9])
40–49	75(33.6)	42/75 (56)	8 (10.7 [5–20.5])	37 (49.3 [37.7–61])	11 (14.7 [7.9–25.2])
50–77	44(19.7)	27/44 (61.4)	4 (9.1 [3–22.6])	20 (45.5 [30.7–61])	8 (18.2 [8.7–33.2])
Median (IQR) 41 (27.5–54.5)					
CD4 cell count, cells/mm^3^					
<200	35 (15.7)	24/35 (68.6)	8 (22.9 [11–40.6])	20 (57.1 [39.5–73.2)	8 (22.9 [11–40.6])
[200–499]	127 (57)	88/127 (69.3)	22 (17.3 [11.4–25.3])	62 (48.8 [40–57.8])	22 (17.3 [11.4–25.3])
≥500	61 (27.3)	17/61 (27.9)	4 (6.6 [2.1–16.7])	11 (18 [9.8–30.4])	2 (3.3 [0.6–12.4])
Median (IQR) 419 (93–745)					
ART regimen					
AZT − 3TC − EFV	94 (42.2)	54/94 (57.4)	15 (16 [9.5–25.3])	40 (42.6 [32.5–53.2])	16 (17 [10.3–26.5])
TDF − 3TC − EFV	13 (5.8)	8/13 (61.5)	2 (15.4 [2.7–46.3])	8 (61.5 [32.3–84.9])	1 (7.7 [0.4–38])
AZT + 3TC + NVP	21 (9.4)	10/21 (47.6)	4 (19 [6.3–42.6])	5 (23.8 [9.1–47.5])	2 (9.5 [1.7–31.8])
TDF + FTC + EFV	54 (24.2)	31/54 (57.4)	7 (13 [5.8–25.5])	22 (40.7 [27.9–54.9])	7 (13 [5.8–25.5])
Others	41 (18.4)	26/41 (63.4)	6 (14.6 [6.1–29.9])	18 (44 [28.8–60.1])	6 (14.6 [6.1–30])

3TC: lamivudine, EFV: efavirenz, TDF: tenofovir, AZT: zidovudine, FTC: emtricitabine, NVP: nevirapin.

**Table 2 viruses-16-00085-t002:** Predictive model explaining the occurrence of opportunistic infections.

Model No. (Rank)	Fixed Effects			Df	ΔAIC	Akaike Weight
	Intercept	CD4 Rate	Sex			
4 (1)	1.534	−0.0031	+	3	0	0.587
2 (2)	1.726	−0.0032		2	0.71	0.413

## Data Availability

The data that support the findings of this study are available on request from the corresponding author.

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
