# Peer review of "Risk Factors Associated with Opportunistic Infections among People Living with HIV/AIDS and Receiving an Antiretroviral Therapy in Gabon, Central Africa"

_viruses, 2024, doi:10.3390/v16010085_

Round 1

Reviewer 1 Report

Comments and Suggestions for Authors

Comments and Suggestions for Authors (with examples/use cases and justification):

  1. Introduction: Provide a more detailed context on the state of HIV/AIDS treatment in Gabon.
    • Example/Use Case: Lines 72-75 discuss Gabon's ART  guidelines based on WHO recommendations.
    • Justification: Understanding the broader context of HIV/AIDS treatment in Gabon can provide valuable background information, enabling readers to appreciate the importance and relevance of the study's findings.
  2. Methods: Elaborate on the sampling strategy and clarify the statistical methods used. Consider addressing potential confounders.
    • Example/Use Case: While lines 165-170 detail the methods, it lacks clarity on how participants were specifically chosen and what exact statistical tests were used.
    • Justification: A clear and detailed methods section is essential for other researchers who may wish to replicate the study or verify its results. It's important to understand if there was any bias in participant selection and how the data was analyzed. Addressing potential confounders ensures that the research findings are robust and less likely to be criticized for overlooking key factors.
  3. Results: Enhance clarity by potentially including tables or figures for key findings.
    • Example/Use Case: The results between lines 200-260 contain critical data on the prevalence of OIs, but there's a lack of visual representation to ease understanding.
    • Justification: Tables and figures provide a visual representation that can help readers quickly grasp key findings and trends. It also breaks up text-heavy sections, making the paper more reader-friendly.
  4. Language: Simplify complex sentences and ensure consistency throughout the manuscript.
    • Example/Use Case: Sentences like the one in lines 294-297 about the role of CD4 T-cells could be simplified for clarity.
    • Justification: Clear and concise language ensures that a wider audience can comprehend the research's significance and findings. Complex or convoluted sentences can obfuscate the research's core message.
  5. Conclusion: It might be beneficial to expand on the conclusions, linking them more explicitly to the broader implications of the study.
    • Example/Use Case: Lines 316-327 summarize the study's findings but could delve deeper into the broader implications, especially in the context of Gabon's healthcare system and recommendations for future actions.
    • Justification: A comprehensive conclusion helps readers understand the broader significance of the study, its implications for the field, and potential next steps.
Comments on the Quality of English Language

Minor edit of language needed

Author Response

Reviewer 1

Comments and Suggestions for Authors (with examples/use cases and justification):

  1. Introduction: Provide a more detailed context on the state of HIV/AIDS treatment in Gabon.
    • Example/Use Case: Lines 72-75 discuss Gabon's ART guidelines based on WHO recommendations.
    • Justification: Understanding the broader context of HIV/AIDS treatment in Gabon can provide valuable background information, enabling readers to appreciate the importance and relevance of the study's findings.

Response:

We thank the reviewer and more details have been provided;

Additional information (a paragraph) related to this request had been included about Gabon’s ART guidelines based on WHO recommendations (2 added references) << Treatment administration in this country follows WHO recommendations for re-source-limited settings/countries………. (INSTI)-based first-line regimens [15]. >>. Please see lines 70-75

  1. Methods: Elaborate on the sampling strategy and clarify the statistical methods used. Consider addressing potential confounders.
    • Example/Use Case: While lines 165-170 detail the methods, it lacks clarity on how participants were specifically chosen and what exact statistical tests were used.
    • Justification: A clear and detailed methods section is essential for other researchers who may wish to replicate the study or verify its results. It's important to understand if there was any bias in participant selection and how the data was analysed. Addressing potential confounders ensures that the research findings are robust and less likely to be criticized for overlooking key factors.

Response:

As recommended by the reviewer, the Materials and Methods section has been improved by providing more details on the cohort of patients recruited in our study <<Patients were recruited between June 2018 to September 2019 to perform……….. filed at the CTA of Franceville>>. Please refer to Lines 93-97

In addition, a figure (Figure 1) illustrating the timeline and the process carried out (medical examinations of viral load, CD4 counts and search for OIs) << Figure 1: Schematic representation of timeline of the study. Progress in carrying out examinations………periods>>. Please refer to Lines 104-107

  1. Results: Enhance clarity by potentially including tables or figures for key findings.
    • Example/Use Case: The results between lines 200-260 contain critical data on the prevalence of OIs, but there's a lack of visual representation to ease understanding.
    • Justification: Tables and figures provide a visual representation that can help readers quickly grasp key findings and trends. It also breaks up text-heavy sections, making the paper more reader-friendly.

Response:

The concern was evaluated and more information was added to elucidate the key findings already contained the manuscript. In the materials and methods, a whole section was added (2.5. Occurrence of opportunistic infections) (Lines 149-159) as well as more information << As a consequence of the AIC change analysis, only the variables (2 factors)…… original seven explanatory variables>> in order to improve the understanding of the results section, specifically Table 2. The units in Table 1 have also been corrected. Please refer to Lines 165-167.

  1. Language: Simplify complex sentences and ensure consistency throughout the manuscript.
    • Example/Use Case: Sentences like the one in lines 294-297 about the role of CD4 T-cells could be simplified for clarity.
    • Justification: Clear and concise language ensures that a wider audience can comprehend the research's significance and findings. Complex or convoluted sentences can obfuscate the research's core message.

Response:

The concern was evaluated and English editing was considered throughout the manuscript, as well as consistency was improved by adding information to elucidate the key findings and issues. All changes are indicated in red in the manuscript.

  1. Conclusion: It might be beneficial to expand on the conclusions, linking them more explicitly to the broader implications of the study.
    • Example/Use Case: Lines 316-327 summarize the study's findings but could delve deeper into the broader implications, especially in the context of Gabon's healthcare system and recommendations for future actions.
    • Justification: A comprehensive conclusion helps readers understand the broader significance of the study, its implications for the field, and potential next steps.

Response:

The comment was considered and a paragraph related to the broader implications and significance of the study was added at the end of section of discussion, and before the conclusion of the manuscript << Therefore, the necessity of improving the management ………. those specific groups of PLWHA>>. Please refer to Lines 334-342

Reviewer 2 Report

Comments and Suggestions for Authors

In this manuscript, the authors have undertaken a comprehensive analysis of risk factors associated with opportunistic infections among individuals living with HIV/AIDS and receiving antiretroviral therapy in Gabon. The study provides valuable insights into the clinical manifestations and potential challenges faced by this specific patient population. However, there are areas in the manuscript that would benefit from additional clarifications, and some language and formatting issues need to be addressed for clarity.

Specific Comments:

1.      Table Clarifications: There seems to be an inconsistency in the data presented in Table 1. The unit of Virological OIs is not consistent with the other two OIs and is confusing. In addition, it is not clear what are the seven factors used to fit into the model in table 2?

2.      Figure: I would consider modifying fig 1 by taking out the HIV part as this is distracting.  What’s more, by saying “tri-infections of HIV, bacteria and parasites were the most prevalent”, I bet the author is comparing among the 3 tri-infection combinations, right?

3.      I would suggest providing some figures presenting the analysis described in the 1st paragraph under section 3.4, which would allow audiences to understand more intuitively.

Author Response

Reviewer 2

In this manuscript, the authors have undertaken a comprehensive analysis of risk factors associated with opportunistic infections among individuals living with HIV/AIDS and receiving antiretroviral therapy in Gabon. The study provides valuable insights into the clinical manifestations and potential challenges faced by this specific patient population. However, there are areas in the manuscript that would benefit from additional clarifications, and some language and formatting issues need to be addressed for clarity.

Specific Comments:

  1. Table Clarifications: There seems to be an inconsistency in the data presented in Table 1. The unit of Virological OIs is not consistent with the other two OIs and is confusing. In addition, it is not clear what are the seven factors used to fit into the model in table 2?

Response:

Indeed, the Table 1 had several errors and they have been corrected throughout.

Also, the clarity concern has been evaluated and addressed with more details added to elucidate the key findings already contained the manuscript. In the materials and methods section, a whole section was added (2.5. Occurrence of opportunistic infections) (Lines 149-159)  as well as more information << As a consequence of the AIC change analysis, only the variables (2 factors)…… original seven explanatory variables>> in order to improve the understanding of the results section, specifically Table 2. Please refer to Lines 165-167.

  1. Figure: I would consider modifying fig 1 by taking out the HIV part as this is distracting.  What’s more, by saying “tri-infections of HIV, bacteria and parasites were the most prevalent”, I bet the author is comparing among the 3 tri-infection combinations, right?

Response:

The Figure 1 has now become Figure 2 after adding a new one. We strongly believe that HIV should be part of the illustration, a removal will forfeit the intended purpose of the authors. The tri-infections refers to the comorbidity made up of the 3 conditions/infections. The statement has been clarified and the word comorbidity included.

  1. I would suggest providing some figures presenting the analysis described in the 1stparagraph under section 3.4, which would allow audiences to understand more intuitively.

Response:

The concern was evaluated and more information was added to elucidate the key findings already contained the manuscript. In the materials and methods section, a whole paragraph was added (2.5. Occurrence of opportunistic infections) (Lines 149-159) as well as more information << As a consequence of the AIC change analysis, only the variables (2 factors)…… original seven explanatory variables>> in order to improve the understanding of the results section. We hope that is suitable to improve the clarity of the analysis and obtained results. Please refer to Lines 165-167

Reviewer 3 Report

Comments and Suggestions for Authors

Augustin et al retrospectively reported the incidences of opportunistic infections (OIs) in a cohort of Gabon that was infected with HIV and received ART. Authors found that TB, HBV/HCV, and oral candidiasis were the top OIs. They also identified male and low CD4 T cell count were the major risk factors of OIs. The topic could be of great interest to the public and manuscript was well written. However, the sample size was limited and some data were missing. The observation was limited.

1.     I have some difficulty understanding the population of the cohort. My understanding is that these patients were diagnosed with HIV from 2017~2019 and the data of CD4 count provided in the manuscript were at the diagnosis. And during January to June 2020, these patients were diagnosed with other OIs.

The OIs developed quite quickly compared to general AIDS patients, likely due to the late diagnosis of HIV (data not provided), poor adherence of ART (not sure about this cohort), and less efficacy of first-line ART. Most importantly, the viral load data at both diagnosis of HIV and OIs were not provided which could be directly associated to CD4 count as well as the incidence of OIs. The high incidence of OIs across all the age ranges and different regimens indicated that HIV was likely not suppressed during the treatment course which undermines the conclusions drawn from the study.

2.     In Table 1, the Median of CD4 cell count and ART regimen were mislabeled.

Author Response

Reviewer 3

Augustin et al retrospectively reported the incidences of opportunistic infections (OIs) in a cohort of Gabon that was infected with HIV and received ART. Authors found that TB, HBV/HCV, and oral candidiasis were the top OIs. They also identified male and low CD4 T cell count were the major risk factors of OIs. The topic could be of great interest to the public and manuscript was well written. However, the sample size was limited and some data were missing. The observation was limited.

  1. I have some difficulty understanding the population of the cohort. My understanding is that these patients were diagnosed with HIV from 2017~2019 and the data of CD4 count provided in the manuscript were at the diagnosis. And during January to June 2020, these patients were diagnosed with other OIs.

The OIs developed quite quickly compared to general AIDS patients, likely due to the late diagnosis of HIV (data not provided), poor adherence of ART (not sure about this cohort), and less efficacy of first-line ART. Most importantly, the viral load data at both diagnosis of HIV and OIs were not provided which could be directly associated to CD4 count as well as the incidence of OIs. The high incidence of OIs across all the age ranges and different regimens indicated that HIV was likely not suppressed during the treatment course which undermines the conclusions drawn from the study.

Response

The reviewer clearly understood the patient recruitment process in this study. More details have changed or adjusted in the Materials and Methods section. It has been improved by providing more details on the cohort of patients recruited in our study <<Patients were recruited between June 2018 to September 2019 to perform……….. filed at the CTA of Franceville. Please refer to Lines 93-97

In addition, a figure (Figure 1) illustrating the timeline and the process carried out (medical examinations of viral load, CD4 counts and search for OIs) << Figure 1: Schematic representation of timeline of the study. Progress in carrying out examinations………periods>>. Please refer to Lines 104-107

  1. In Table 1, the Median of CD4 cell count and ART regimen were mislabelled.

Response:

Indeed, the Table 1 had several errors and they have been corrected throughout.

Round 2

Reviewer 1 Report

Comments and Suggestions for Authors

The manuscript is much improved but the authors will need the services of a English Language editor to finetune the language of this manuscript before it can be published. There are several run on sentences, punctuation issues, several issues with subject verb agreement and passive sentences. Few have been identified below but the list is not exhaustive

instead of " Biological 32

and socio-demographic characteristics were examined concerning OIs through both an univariate 33

analysis via Fisher's exact tests or chi² (χ²), and a multivariate analysis via logistic regression.", it should be "Biological and socio-demographic characteristics were examined in relation to OIs using both univariate analysis through Fisher's exact tests or chi-squared (χ²), and multivariate analysis via logistic regression."

why is 13-29 missing in this cohort:"The mean age was 40 (38.6; 41.85), with individuals ranging from 2 to 77 years old. 36

The study cohort was classified into five age groups (2 to 12, 20 to 29, 30 to 39, 40 to 49, and 50 to 77 37

years old),"

Line 48-54, availability may be appropriate word than vulgarization

Regarding the second statement, there's a grammatical issue with subject-verb agreement, and the list should be parallel. Here's the refined version:

"Factors related to late ART initiation, discontinuation, poor adherence, low CD4 T lymphocyte counts, inadequate virological monitoring, lack of isoniazid preventive therapy, as well as gender, age, place of residence, functional status, disclosure status, and nutritional status, have been associated with the emergence of opportunistic infections (OIs) among adults living with HIV/AIDS post-ART initiation."

Line 57- ART-administered adults?

Line 59- you dont need from here "from 33.6 to 88.44% in Ethiopia"

Line 71-resource-limited and not re-source-limited

line 80-81- chnage to same thing as in the abstract, "Outpatient or Ambula- 80

tory Treatment Centre (CTA) of Franceville, "

Line 76-remove "in" in this sentence"Studies have been conducted on HIV in co-infection with other pathogens"

Comments on the Quality of English Language

The manuscript is much improved but the authors will need the services of a English Language editor to finetune the language of this manuscript before it can be published. There are several run on sentences, punctuation issues, several issues with subject verb agreement and passive sentences. Few have been identified below but the list is not exhaustive

instead of " Biological 32

and socio-demographic characteristics were examined concerning OIs through both an univariate 33

analysis via Fisher's exact tests or chi² (χ²), and a multivariate analysis via logistic regression.", it should be "Biological and socio-demographic characteristics were examined in relation to OIs using both univariate analysis through Fisher's exact tests or chi-squared (χ²), and multivariate analysis via logistic regression."

why is 13-29 missing in this cohort:"The mean age was 40 (38.6; 41.85), with individuals ranging from 2 to 77 years old. 36

The study cohort was classified into five age groups (2 to 12, 20 to 29, 30 to 39, 40 to 49, and 50 to 77 37

years old),"

Line 48-54, availability may be appropriate word than vulgarization

Regarding the second statement, there's a grammatical issue with subject-verb agreement, and the list should be parallel. Here's the refined version:

"Factors related to late ART initiation, discontinuation, poor adherence, low CD4 T lymphocyte counts, inadequate virological monitoring, lack of isoniazid preventive therapy, as well as gender, age, place of residence, functional status, disclosure status, and nutritional status, have been associated with the emergence of opportunistic infections (OIs) among adults living with HIV/AIDS post-ART initiation."

Line 57- ART-administered adults?

Line 59- you dont need from here "from 33.6 to 88.44% in Ethiopia"

Line 71-resource-limited and not re-source-limited

line 80-81- chnage to same thing as in the abstract, "Outpatient or Ambula- 80

tory Treatment Centre (CTA) of Franceville, "

Line 76-remove "in" in this sentence"Studies have been conducted on HIV in co-infection with other pathogens"

Author Response

Reviewer 1

Comments and Suggestions for Authors

The manuscript is much improved but the authors will need the services of a English Language editor to finetune the language of this manuscript before it can be published. There are several run on sentences, punctuation issues, several issues with subject verb agreement and passive sentences. Few have been identified below but the list is not exhaustive.

We appreciate the author remark concerning the English Language, and we thank him for the sentences identified below. The manuscript was modified accordingly. We finally sent the paper to be read by a scientist, English borned. The new edited words and sentences are highlighted in blue, in the manuscript.

instead of " Biological and socio-demographic characteristics were examined concerning OIs through both an univariate analysis via Fisher's exact tests or chi² (χ²), and a multivariate analysis via logistic regression.", it should be "Biological and socio-demographic characteristics were examined in relation to OIs using both univariate analysis through Fisher's exact tests or chi-squared (χ²), and multivariate analysis via logistic regression."

We thank the reviewer for this suggestion. This paragraph was modified again during the English Language editing. Please see lines 32-34.

why is 13-29 missing in this cohort:"The mean age was 40 (38.6; 41.85), with individuals ranging from 2 to 77 years old. The study cohort was classified into five age groups (2 to 12, 20 to 29, 30 to 39, 40 to 49, and 50 to 77 37years old),"

We thank the author for this question. In the study cohort, we did not register patients aged to 13, 14, 15, 16, 17, 18 and 19 years. That is why after the first age group (2 to 12), the second one was 20 to 29.

Line 48-54, availability may be appropriate word than vulgarization

As requested by the reviewer, « vulgarization » was replaced by « availability »

Regarding the second statement, there's a grammatical issue with subject-verb agreement, and the list should be parallel. Here's the refined version: 

"Factors related to late ART initiation, discontinuation, poor adherence, low CD4 T lymphocyte counts, inadequate virological monitoring, lack of isoniazid preventive therapy, as well as gender, age, place of residence, functional status, disclosure status, and nutritional status, have been associated with the emergence of opportunistic infections (OIs) among adults living with HIV/AIDS post-ART initiation."

We thank again the reviewer for his help to improve our paper. This refined version provided by him was revisited by the English language editor in the new version of the manuscript.

Line 57- ART-administered adults?

This group of words was totally replaced during the English Language editing.

Line 59- you dont need from here "from 33.6 to 88.44% in Ethiopia"

As requested by the reviewer, « from » was deleted.

Line 71-resource-limited and not re-source-limited

We edited « resource-limited », accordingly with the reviewer remark

line 80-81- chnage to same thing as in the abstract, "Outpatient or Ambulatory Treatment Centre (CTA) of Franceville, "

We thank the reviewer for this remark. « Outpatient Treatment Centre (CTA) » is finally the appropriate use in the manuscript.

Line 76-remove "in" in this sentence"Studies have been conducted on HIV in co-infection with other pathogens"

We thank the reviewer for this request. Finally, this sentence was rewritten during the English Language editing.

Comments on the Quality of English Language

Reviewer 3 Report

Comments and Suggestions for Authors

I am fine with the revision.

Author Response

We thank the reviewer for his final decision: "I am fine with the revision"